# Comprehensive Evaluation of 202 Cotton Varieties (Lines) and Their Physiological Drought Resistance Response During Seedling Stage

**DOI:** 10.3390/plants14121770

**Published:** 2025-06-10

**Authors:** Jiazila Baha, Wenhong Liu, Xiaoman Ma, Yage Li, Xiaohong Zhao, Xue Zhai, Xinchuan Cao, Weifeng Guo

**Affiliations:** 1College of Agriculture, Tarim University, Alar 843300, China; jiazilabaha@163.com (J.B.); maxiaoman2024@163.com (X.M.); liyage010904@163.com (Y.L.); zhaoxiaohong0109@163.com (X.Z.); zhaixue233@163.com (X.Z.); cxczky@163.com (X.C.); 2Key Laboratory of Genetic Improvement and Efficient Production for Specialty Crops, Arid Southern Xinjiang of Xinjiang Corps, Alar 843300, China; 3Asset Management Divisio, Shihezi University, Shihezi 832003, China

**Keywords:** cotton, seedling stage, drought resistance, entropy weight method, comprehensive evaluation

## Abstract

To identify seedling traits closely associated with drought resistance and to screen for drought-tolerant germplasm, 202 cotton varieties (lines) were evaluated under controlled indoor conditions using a nutrient soil cultivation method. Seedling-stage traits measured included plant height, cotyledon node diameter, true leaf number, chlorophyll content, and fresh and dry biomass of both shoots and roots. Drought resistance was assessed using drought resistance coefficients for each trait, followed by descriptive statistics, principal component analysis (PCA), partial correlation analysis, and comprehensive evaluation via the entropy weight method. PCA and partial correlation analysis revealed that plant height, cotyledon node diameter, aboveground fresh weight, and underground fresh weight were strongly associated with drought resistance at the seedling stage. The comprehensive drought resistance index (D-value) classified the 202 cotton lines into four categories: highly drought-resistant, moderately drought-resistant, drought-sensitive, and highly drought-sensitive. Physiological assays indicated that malondialdehyde (MDA) content in drought-resistant lines first increased and then declined with prolonged drought stress, while it continued to increase in sensitive lines. In contrast, proline (Pro) content and superoxide dismutase (SOD) activity increased steadily in drought-resistant lines but showed negligible changes in sensitive lines. These four morphological traits and three physiological indicators represent reliable criteria for evaluating drought resistance in cotton seedlings. Four highly drought-resistant and thirteen moderately drought-resistant lines were identified, providing valuable germplasm for genetic improvement of drought tolerance in cotton.

## 1. Introduction

Cotton is an important economic crop in Xinjiang, which is of great significance for increasing the income of local farmers and is also an important part of China’s economic development and food security. Due to the increasing scarcity of global water resources, drought and semi-arid are the main climatic features for the inland areas of northwest China, with Xinjiang having the largest arid area. Therefore, water scarcity has become a global agricultural production problem [1]. The cotton production in Xinjiang plays a crucial role and important task in maintaining the safety of cotton production in China, meeting the needs of the cotton textile industry and promoting the economic development of Xinjiang [2]. In order to alleviate the adverse effects of drought and water shortage on cotton production, it is necessary to accelerate the cultivation of high-yield and drought-resistant varieties, which is an effective way to alleviate drought stress [3]. Therefore, exploring the drought resistance of cotton germplasm and cultivating highly drought-resistant cotton varieties are of great significance for the development of Xinjiang’s cotton industry [4].

The comprehensive evaluation value (D) of drought resistance is the weight value of multiple traits, which integrates the drought resistance coefficients of different traits and effectively reflects the comprehensive performance of crops under drought stress [5,6]. In the study of drought resistance in different varieties, scholars at home and abroad have selected indicators and drought-resistant varieties from different perspectives, provided evaluation methods, and evaluated drought resistance [7,8]. Drought resistance indicators are widely used in the study of drought resistance in common crops, such as cotton, foxtail millet, and oat [9,10,11]. The entropy method is an objective weighting method, which determines weights based on the amount of information contained in each indicator, making the results objective. The core concept of this method is information entropy, which can effectively reflect the degree of variation of indicators [12].

The antioxidant mechanism of crops inhibits oxidative reactions by producing superoxide dismutase (SOD), thereby reducing cell damage, such as in wheat [13] and cassava [14]. The SOD activity of cotton varieties with strong drought resistance increases with prolonged drought treatment time [15], and the increase in antioxidant enzyme activity can enhance crop comprehensive resistance [16]. The accumulation of proline (Pro) and malondialdehyde (MDA) in cotton is a characteristic associated with drought resistance [17,18]. Proline (Pro), as an osmotic regulator, and changes in its content can reflect the crop’s resistance to drought stress. When plants are subjected to drought stress, the osmotic stress acts on the cells, and plants regulate the osmotic potential inside the cells by accumulating osmoregulatory substances to maintain water balance in the body. Increasing MDA content in plants subjected to drought stress, this is due to the increase in reactive oxygen species (ROS) leading to increased oxidative stress in the plant thereby generating membrane lipid peroxidation, which exhibits the extent of oxidative damage to the cells [19]. Studies have shown that plants can alleviate the effects of stress by increasing cell membrane permeability, reducing plasma membrane peroxidation, and enhancing antioxidant enzyme activity [20].

Therefore, this experiment adopted an indoor nutrient soil cultivation method, using 202 cotton varieties (lines) as experimental materials, and measured eight traits during the seedling stage under drought stress and normal irrigation conditions. Using descriptive statistical analysis, principal component analysis, and correlation analysis, combined with the entropy weight method, drought resistance evaluation is carried out to screen seedling traits and drought-resistant variety resources closely related to drought resistance. SOD, Pro and MDA indicators were measured for five drought-resistant and five drought-sensitive varieties. The main objective of this study was to comprehensively evaluate drought resistance in cotton seedlings using morphological traits and physiological indices under drought stress, providing a theoretical basis for genetic improvement of drought resistance in cotton.

## 2. Results

### 2.1. Analysis of Drought Resistance of 202 Cotton Varieties (Lines)

#### 2.1.1. Statistical Analysis of Drought Resistance Coefficient (Dc)

Statistical analysis was conducted on the drought resistance coefficients of eight traits (Table 1). The average values of plant height, cotyledon node diameter, true leaf number, aboveground fresh weight, aboveground dry weight and underground fresh weight were less than 1, indicating that six traits were inhibited by drought stress. The average values of underground dry weight and chlorophyll content were greater than 1, indicating that two traits showed an increasing trend under drought stress. There is a significant difference in the average values of eight traits, indicating that different traits are affected to varying degrees by drought, with aboveground fresh weight being the most affected. The coefficient of variation of the eight traits ranges from 21.15% to 109.64%, with the smallest standard deviation for plant height, the largest standard deviation for underground dry weight, and the highest coefficient of variation for underground fresh weight, which is 109.64%. This indicates that the eight traits have diverse phenotypes under drought stress.

#### 2.1.2. ANOVA and Correlation Analysis of Eight Traits

Eight traits showed significant difference among 202 varieties (lines) (Table 2). The correlation analysis of the drought resistance coefficient of eight traits showed that there was a significant positive correlation between plant height and the diameter of cotyledon node, number of true leaves, aboveground fresh weight, aboveground dry weight, underground dry weight. There is a significant positive correlation between the diameter of the cotyledon node and aboveground fresh weight, aboveground dry weight. There is a significant positive correlation between the number of true leaves and aboveground fresh weight, aboveground dry weight, and underground dry weight. There is a significant positive correlation between aboveground fresh weight and underground fresh weight, aboveground dry weight, and underground dry weight. There is a significant positive correlation between underground fresh weight and underground dry weight. There is a significant positive correlation between aboveground dry weight and underground dry weight. However, there is a significant negative correlation between chlorophyll content and plant height, as well as cotyledon node diameter. There is a significant negative correlation between chlorophyll content and aboveground fresh weight and aboveground dry weight (Figure 1). In order to comprehensively evaluate the drought resistance of cotton seedlings, in-depth analysis is needed.

#### 2.1.3. Principal Component Analysis of Drought Resistance Coefficient for Eight Traits

Principal component analysis (PCA) was conducted on eight traits of 202 varieties (lines) (Table 3; Figure 2), and the factor load and contribution rate of eight traits were obtained. The eigenvalues of three principal components PC1, PC2 and PC3 were greater than 1, and the cumulative contribution rate of the three principal components was 65.57%. According to the eigenvalue and factor load value, the eigenvalue of PC1 is 3.10, with a contribution rate of 38.77%. The load values of plant height (0.50), aboveground fresh weight (0.49) are relatively large. The eigenvalue of PC2 is 1.107, with a contribution rate of 13.83%, and the underground fresh weight load value (0.68) is the highest. The eigenvalue of PC3 is 1.04, with a contribution rate of 12.97%, and the load value of cotyledon node diameter is the highest (0.58). Principal component analysis shows that plant height, aboveground fresh weight, underground fresh weight and cotyledon node diameter can be used as drought resistance evaluation indicators for cotton seedlings.

#### 2.1.4. Partial Correlation Analysis Between Drought Resistance Coefficient and D-Value of Eight Traits

Further analysis was conducted on the partial correlation between the drought resistance coefficient and D-value of eight traits, and the drought resistance coefficient and D-value of all eight traits were significantly positively correlated. Based on the analysis of principal components and partial correlation, plant height, cotyledon node diameter, aboveground fresh weight and underground fresh weight are significantly correlated with the drought resistance of cotton seedlings. These four traits can be used as important indicators for identifying drought resistance in cotton seedlings (Table 4).

#### 2.1.5. D-Value Analysis of 202 Cotton Varieties (Lines)

Based on the drought resistance coefficients of eight traits of 202 cotton varieties (lines), the entropy weight method was used to calculate the comprehensive drought resistance D-value of each variety (line). The D-value ranges from 0.07 to 0.63, with an average of 0.20. The larger the D-value, the stronger the drought resistance, while the smaller the D-value, the weaker the drought resistance. According to the D-value, 202 cotton varieties (lines) are classified into four levels: D ≥ 0.40 is the highly drought-resistant type, 0.30 ≤ D < 0.40 is the moderately drought-resistant type, 0.20 ≤ D < 0.30 is the drought-sensitive type, and D < 0.20 is the highly drought-sensitive type. J206-5, jiumian20, TD1 and zhongmiansu27 are in the highly drought-resistant type. A total of 13 materials (xinluzhong87, sumian7381, TD2, etc.) fall into the moderately resistant type. A total of 71 materials (TD3, TD4, xinkenK73, etc.) fall into the drought-sensitive type. A total of 114 materials (23NJH02, xinluzao12, TD17, etc.) fall into the highly drought-sensitive type (Table 5). A total of 17 varieties (lines) can provide drought-resistant germplasm resources for cotton breeding.

Detailed information about standardised treatment values, D-values, drought tolerance classification, and cluster analysis chart of drought resistance coefficients is provided in Appendix A.

### 2.2. Analysis of Physiological Indicators Under Drought Stress

#### 2.2.1. Analysis of SOD (Drought Stress/Control) Under Drought Stress

There was a significant difference in SOD between drought-resistant and drought-sensitive materials under drought stress. There were significant differences in SOD among the 5 time points for drought-resistant and sensitive materials. There were significant differences in SOD between drought-resistant and sensitive materials at the same time point (Table 6). The SODs of drought-resistant materials were higher than those of drought-sensitive materials on 13d, 16d and 19d, with the highest SOD activity of 3.51 U/mg recorded on day 19d. The SOD of drought-resistant materials showed a significant increase trend with the prolongation of drought treatment time, while the SOD of drought-sensitive materials showed a decreasing trend with the prolongation of drought treatment time (Figure 3).

#### 2.2.2. Pro (Drought Stress/Control) Analysis Under Drought Stress

There was a significant difference in Pro content between drought-resistant and drought-sensitive materials under drought stress. There were significant differences in the Pro content of drought-resistant materials among the five time points, while there were no differences in the Pro content of drought-sensitive materials among the five time points. There were significant differences in Pro content between drought-resistant and drought-sensitive materials at the same time point (Table 7). The Pro content values of drought-resistant materials at 13d, 16d and 19d were higher than those of drought-sensitive materials, with the highest value of 53.90 μg/g FW at 19d. The Pro content of drought-resistant materials showed a significant increase trend with the prolongation of drought treatment time, while the Pro of drought-sensitive materials did not change significantly with the prolongation of drought treatment time (Figure 4).

#### 2.2.3. MDA (Drought Stress/Control) Analysis Under Drought Stress

Under drought stress, there was no significant difference in MDA content between drought-resistant and drought-sensitive materials. There was significant difference in MDA between drought-resistant and drought-sensitive materials at five time points. There was significant difference in MDA between drought-resistant and drought-sensitive materials at the same time point (Table 8). The MDA content of drought-resistant materials at 16d and 19d were lower than those of drought-sensitive materials, with the highest MDA observed in drought-resistant materials at 10d. MDA levels initially increased and then declined in drought-resistant materials, while continuing to rise in sensitive materials (Figure 5).

## 3. Discussion

### 3.1. Screening and Comprehensive Evaluation of Drought Resistance Morphological Traits in Cotton Seedling

The drought resistance of cotton seedlings is a complex quantitative trait, and reasonable indicators are pivotal to drought resistance identification. With the in-depth study of crop drought resistance, different drought resistance evaluation methods have been proposed. The comprehensive analysis of crop drought resistance mainly adopts principal component analysis, membership function method, factor analysis method, and cluster analysis method. The comprehensive evaluation method of multiple traits is used to determine the drought resistance of experimental materials [18,21,22,23].

The variation coefficient of different traits reflects the difference in the response of experimental materials to drought stress. The larger the coefficient of variation, the greater the difference in the response of experimental materials to drought stress, which is the basis for evaluating the drought resistance of experimental materials [24]. The variation coefficient of drought resistance coefficients for various traits in this experiment ranged from 21.15% to 109.64%, with the highest coefficient of variation for underground fresh weight, indicating the diversity of drought stress performance of this trait among the experimental materials (Table 1). When crops are subjected to drought stress, different traits will change, and morphological traits are often used as indicators for identifying drought resistance [25,26]. This experiment showed significant correlation among eight morphological traits through correlation analysis (Table 2; Figure 1). Principal component and partial correlation analysis can describe the different trait characteristics of drought resistance and salt tolerance in experimental materials [27,28,29]. Soil water deficit significantly affected the growth and development of cotton plant height, especially under severe drought conditions [30]. Studies have shown that the plant height and growth rate of cotton were closely related to the degree of deficit irrigation, and there were differences in the sensitivity of plant height and growth rate to water at different growth stages [31]. When cotton at the seedling stage was subjected to moderate drought, there was no significant difference in the inhibition effect of drought stress on plant height growth, and the sensitivity to drought in the early stage of seedling was higher than that in the late stage of seedling [32]. The experimental results show that plant height, cotyledon node diameter, aboveground fresh weight and underground fresh weight are significantly correlated with the drought resistance of cotton seedling. These four traits can be helpful for accelerate the breeding process of new drought resistance cotton varieties, help to explore the internal relationship between crops morphological structure and water stress response, and provide theoretical support for optimizing cultivation and management measures (Table 3; Table 4). Analytic Hierarchy Process (AHP) was proposed by Thomas L. Saaty, an American operations research scientist, in the 1970s [33]. It is a multi-criteria decision-making method to determine the weight of factors by decomposing the hierarchical structure and combining qualitative and quantitative analysis. The APH method is a subjective weighting method, which is based on the knowledge and experience of the evaluator. It is interpretable and systematic, but it has a lack of subjective randomness. Therefore, considering that the AHP method only relies on expert experience and ignores the measured data when determining the index weight, and the entropy weight method can avoid subjective randomness [34], this study uses the entropy weight method to calculate the weight of each traits.

This experiment comprehensively evaluated the drought resistance of 202 cotton varieties (lines) using the entropy weight method, and classified the drought resistance of the experimental materials into four levels based on the D-value (Table 5). J206-5, jiumian20, TD1 and zhongmiansuo27 are highly drought-resistant varieties (lines), while 13 varieties (lines) show moderate drought resistance. These materials can be used as germplasm resources for genetic research and breeding improvement of cotton drought resistance (Table 5).

### 3.2. Characteristics of Physiological Indicators in Cotton Seedling Under Drought Stress

Reports on the effects of drought stress on crop physiological characteristics indicate that drought-resistant crops can maintain high level of antioxidant enzyme activity under drought stress, and there is a positive correlation between SOD activity and antioxidant capacity [35]. In this experiment, the relative values of SOD in drought-resistant varieties at 13d, 16d and 19d were higher than those in drought-sensitive varieties. The drought-resistant varieties showed an increasing trend with prolonged drought treatment time (Figure 3), indicating that SOD can enhance the drought resistance of cotton seedlings during drought stress, which is consistent with the experimental result of Jian Yu [36].

Cotton varieties with strong drought resistance have significantly higher Pro content in their leaves after drought stress compared to varieties with weak drought resistance [37]. The sugarcane variety ‘Xintaitang 22’ was subjected to drought stress. As the drought stress intensified, the content of free amino acids and proline increased, and the increase was significant in the later stage of drought stress, with significant difference compared to the control material [38]. In this experiment, the relative values of Pro in drought-resistant varieties at 13d, 16d and 19d were higher than those in drought-sensitive varieties under drought stress. The drought-resistant varieties showed an increasing trend with the prolongation of drought treatment time, and the highest Pro content was observed at day 19 (Figure 4), indicating that Pro plays an important role in the drought resistance of cotton seedling.

Under environmental stress, crops undergo peroxidation of cell membrane lipid to produce MDA. The higher the intracellular MDA content, the stronger the damage to the cell membrane, which affects the normal physiological response of cells [39,40]. In this experiment, the MDA content of drought-sensitive varieties showed a significant increase trend with prolonged drought treatment time under drought stress, but the MDA content of drought-resistant varieties first increased and then significantly decreased (Figure 5), indicating that drought-resistant varieties can improve their drought resistance by reducing MDA content, which is consistent with the experimental results of Xiaoyu Xie [41].

## 4. Conclusions

Plant height, cotyledon node diameter, aboveground fresh weight and underground fresh weight are important morphological characteristics of drought resistance in cotton seedling stage. SOD, Pro and MDA play important roles in the drought resistance of cotton seedlings. The screened drought-resistant materials can be used as germplasm resource for genetic research and breeding improvement of cotton drought resistance.

## 5. Materials and Methods

### 5.1. Materials

Drought treatment during cotton seedling stage: 202 cotton varieties (lines).

Physiological indicators of cotton seedling stage:

Drought-resistant materials: J206-5, jiumian20, K7, xinshiH16, zhongmiansuo45;

Sensitive drought materials: xinluzhong77, xinluzhong67, xinluzao4, zhaofeng1, jinmian 16.

The materials are provided by the germplasm resource bank of the College of Agriculture, Tarim University.

### 5.2. Methods

Drought stress tests were carried out in the light cultivation room of the College of Agriculture, Tarim University, with a light intensity of 1400 umol/s/m^2^, day and night temperatures of 30 °C and 21 °C, day and night time of 16 h and 8 h, and relative humidity of 60%. Two treatments were set up: drought stress and normal irrigation (CK). Cotton seeds of the same size were sown in nutrient soil using small seedling pots with small holes at the bottom (upper diameter of 12 cm, lower diameter of 10 cm, height of 10 cm), with a sowing depth of 2 cm. Twelve small seedling pots were placed in a large seedling pot with a transparent moisturizing cover, and 250 mL Hoagland nutrient solution was poured into each small pot. After the cotyledons were flattened, the moisturizing transparent cover was removed. The drought treated material was no longer watered with nutrient solution, and CK was watered with 50 mL nutrient solution every 3 days.

Investigation of seedling traits: After 15 days of drought stress, the plant height, cotyledon node diameter, number of true leaves, chlorophyll content, aboveground and underground dry and fresh weight of drought and CK materials were measured. 3 biological replicate experiments were performed for each variety (line).

Measurement of physiological indicators: Based on the comprehensive measurement of drought resistance, 5 drought-resistant varieties and 5 sensitive varieties were selected for control and drought stress treatment, respectively. Leaf sampling times were at 7 days, 10 days, 13 days, 16 days and 19 days. The content of proline (Pro) was determined using the acidic ninhydrin method [42], the activity of superoxide dismutase (SOD) was determined using the nitrogen blue tetrazole photoreduction method [43], and the content of malondialdehyde (MDA) was determined using the thiobarbituric acid method [44]. Three biological replicate experiments were performed for each variety (line).

### 5.3. Comprehensive Evaluation of Drought Resistance

The entropy weight method [12] is used to calculate the weight of each trait and the comprehensive measure of drought resistance for each variety (line).

(1)Calculate the drought resistance coefficient (Dc) according to formula (1):

Dc = drought treatment value of each trait/control value of each trait(1)

(2)Standardization of various traits: Standardize the positive and negative traits according to formulas 2 and 3, respectively.


(2)
$$positive\;traits{\;X}_{ij}=\left[X_{ij}-min\left(X_{ij}\right)\right]/\left[max\left(X_{ij}\right)-min\left(X_{ij}\right)\right]$$



(3)
$${negative\;traits\;X}_{ij}=1-\left[X_{ij}-min\left(X_{ij}\right)\right]/\left[max\left(X_{ij}\right)-min\left(X_{ij}\right)\right]$$


*X_ij_* is the measured value of the *j*-th trait of the *i*-th variety; max (*X_ij_*) represents the maximum value of the *j*-th trait; min (*X_ij_*) represents the minimum value of the *j*-th trait.

(3)Data translation processing: To ensure the validity of the values, add 0.01 to each standardized number.(4)Calculate the proportion (*P_ij_*) of the *j*-th trait of the *i*-th variety (line) according to formula (4):


(4)
$$P_{ij}=\frac{P_{ij}}{\Sigma_{j=1}^{n}P_{ij}}$$


(5)Calculate the entropy value (*e_j_*) of each trait according to formula (5):


(5)
$$e_{j}=-K\;*\sum_{j=1}^{n}\left[P_{ij\;}*\;ln\left(P_{ij}\right)\right]$$


K = 1/ln(*n*), n represents different cotton varieties (lines).

(6)Calculate the information utility value (*d_j_*) of each trait according to formula (6):

*d_j_* = 1 − *e_j_*(6)

(7)Calculate the weight (*W_j_*) of each trait according to formula (7):


(7)
$$w_{j}=\frac{d_{i}}{\sum_{i=1}^{n}d_{i}}$$


(8)Calculate the comprehensive drought resistance measurement value (D) of different cotton varieties (lines) according to formula (8):


(8)
$$D=\sum_{j=1}^{n}X_{ij}W_{j}$$


### 5.4. Data Analysis

Microsoft Excel was used for data organization, SPSS 20.0 software was used for statistical analysis and principal component analysis, and Origin Pro 2024 was used for correlation analysis and plotting.

## Figures and Tables

**Figure 1 plants-14-01770-f001:**
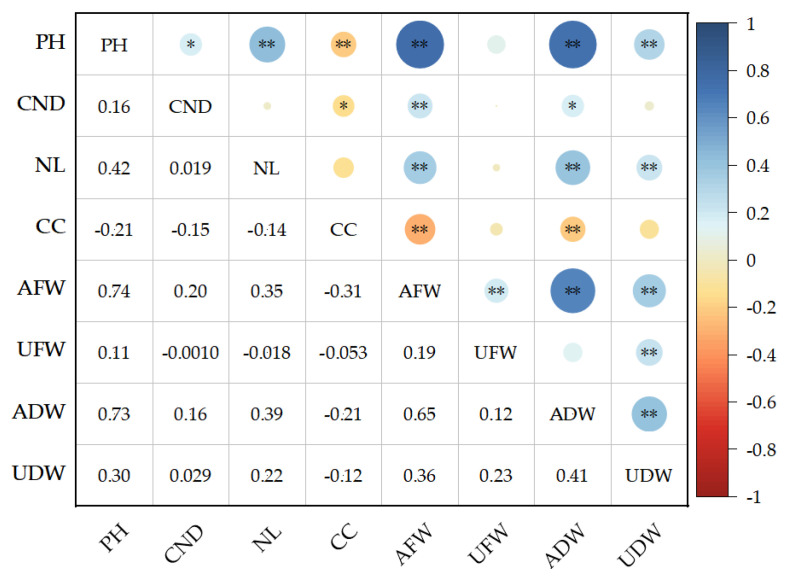
Correlation analysis between eight traits (Pearson correlation coefficient). Note: PH, plant height; CND, cotyledon node diameter; NL, number of true leaves; CC, chlorophyll content; AFW, aboveground fresh weight; UFW, underground fresh weight; ADW, aboveground dry weight; UDW, underground dry weight. * and ** indicate significant correlations at the 0.05 and 0.01 level, respectively.

**Figure 2 plants-14-01770-f002:**
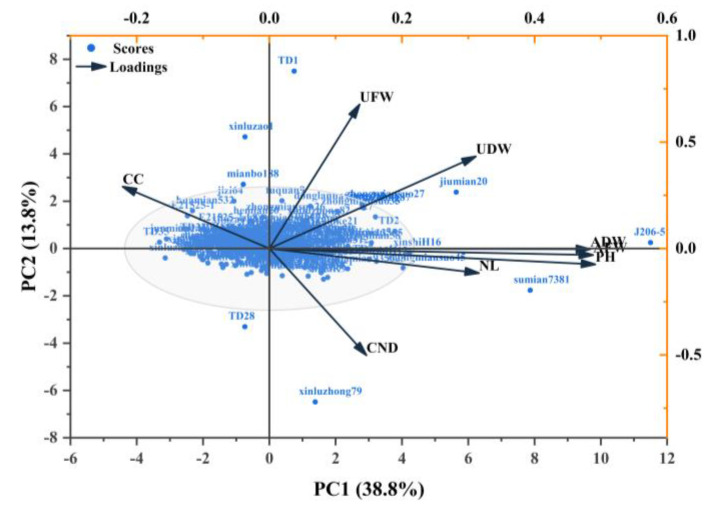
Principal component analysis scatter plot of drought resistance coefficients for 8 traits. PH, plant height; CND, cotyledon node diameter; NL, number of true leaves; CC, chlorophyll content; AFW, aboveground fresh weight; UFW, underground fresh weight; ADW, aboveground dry weight; UDW, underground dry weight.

**Figure 3 plants-14-01770-f003:**
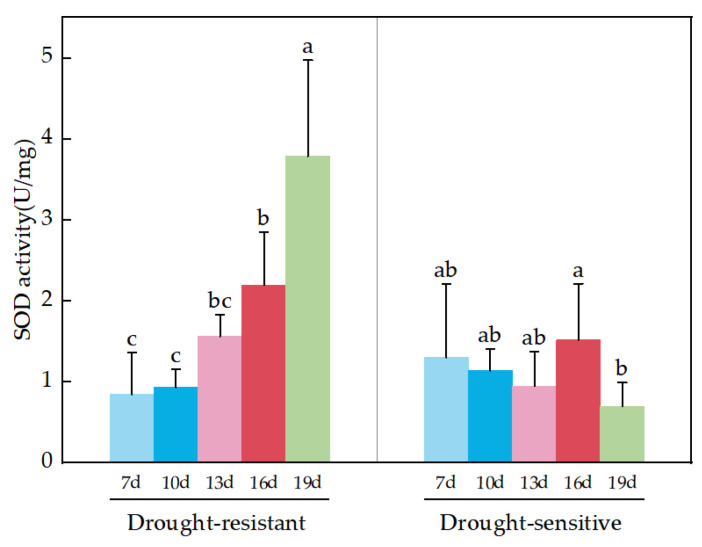
SOD analysis of drought-resistant and drought-sensitive materials under drought stress. Note: (a–c) markers of significance of differences at 0.05 level.

**Figure 4 plants-14-01770-f004:**
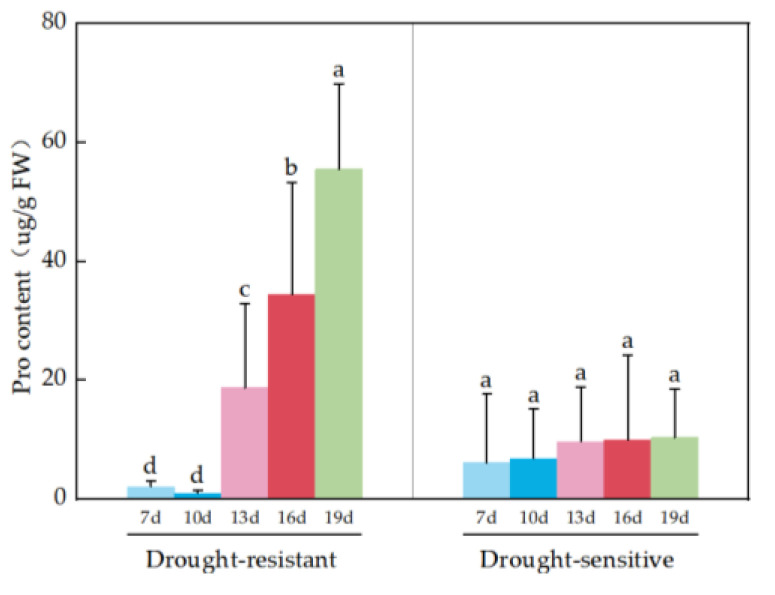
Pro analysis of drought-resistant and drought-sensitive materials under drought stress. Note: (a–d) markers of significance of differences at 0.05 level.

**Figure 5 plants-14-01770-f005:**
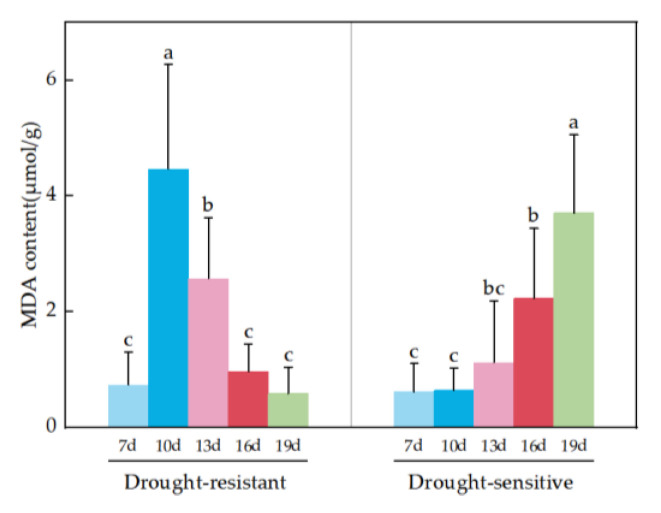
MDA analysis of drought-resistant and drought-sensitive materials under drought stress. Note: (a–c) markers of significance of differences at 0.05 level.

**Table 1 plants-14-01770-t001:** Statistical analysis of drought resistance coefficient of the eight traits.

Traits	Min	Max	Mean ± SD	CV (%)
PH	0.52	1.81	0.86 ± 0.18 c	21.15
CND	0.06	7.03	0.87 ± 0.52 c	59.52
NL	0.00	2.50	0.68 ± 0.29 de	43.19
CC	0.00	2.51	1.17 ± 0.38 a	32.74
AFW	0.25	2.79	0.59 ± 0.25 e	42.92
UFW	0.05	7.82	0.63 ± 0.69 e	109.64
ADW	0.09	1.90	0.73 ± 0.24 d	32.99
UDW	0.00	4.88	1.04 ± 0.85 b	81.38

Note: (a–e) markers of significance of differences at 0.05 level. PH, plant height; CND, cotyledon node diameter; NL, number of true leaves; CC, chlorophyll content; AFW, aboveground fresh weight; UFW, underground fresh weight; ADW, aboveground dry weight; UDW, underground dry weight. The same as below.

**Table 2 plants-14-01770-t002:** Analysis of variance for each trait.

Source of Variation	Square	Degree of Freedom	Mean Square	F Value	*p* Value
Between treatment	74.38	201	0.37	1.47	0.0001
Within treatment	356.66	1414	0.25		
Total variation	431.04	1615			

**Table 3 plants-14-01770-t003:** Principal component analysis of drought resistance coefficients for eight traits.

Traits	Principal Component (PC)		
	PC1	PC2	PC3
Plant Height	** 0.50 **	−0.08	−0.13
Cotyledon Node Diameter	0.15	−0.50	** 0.58 **
Number of True Leaves	0.32	−0.11	−0.49
Chlorophyll Content	−0.22	0.29	−0.40
Aboveground Fresh Weight	** 0.49 **	−0.03	0.05
Underground Fresh Weight	0.14	** 0.68 **	0.48
Aboveground Dry Weight	0.48	0.00	−0.11
Underground Dry Weight	0.31	0.43	0.04
Eigenvalue	3.10	1.10	1.04
Contribution %	38.77	13.83	12.97
Cumulative Contribution %	38.77	52.59	65.57

Note: Red indicates the highest loading of each principal component.

**Table 4 plants-14-01770-t004:** Partial correlation between drought resistance coefficient and D-value of eight traits.

Traits	Comprehensive Evaluation D-Value	*p* Value
Plant height	0.74 **	0.000
Cotyledon Node Diameter	0.60 **	0.000
Number of True Leaves	0.74 **	0.000
Chlorophyll Content	0.73 **	0.000
Aboveground Fresh Weight	0.81 **	0.000
Underground Fresh Weight	0.94 **	0.000
Aboveground Dry Weight	0.53 **	0.000
Underground Dry Weight	0.98 **	0.000

Note: ** indicate significantly partial correlation at the 0.01 level.

**Table 5 plants-14-01770-t005:** Drought resistance levels of 202 cotton varieties (lines) during the seedling stage.

Drought Resistance Level	D-Value Range	Varieties/Lines
highly drought-resistant	0.40~0.63	J206-5, jiumian20, TD1, zhongmiansuo27
moderately drought-resistant	0.30~0.39	xinluzhong87, sumian7381, TD2, donglannaxiangdahua, zhongmiansuo35, K7, xinshiH16, mianbo188, jinke21, fuquan9, zhongmiansuo45, jimian5, xinshiH12
drought -sensitive	0.20~0.29	TD3, TD4, xinkenK73, chuangmian58, chuan2806, xinluzhong82, HN1409, TD5, xinluzao1, TD6, jijiaomian, TD7, zhongmiansuo26, yuanminanxin13305, jinken1441, jizi64, TD8, TD9, meifuchangrong, zhongda4, xiyu1, chuangmian512, huamian935623, zhongzhi2B, zhongmiansuo17, haixingnaiyan10, lv5, xinluzhong79, zhongmiansuo143, ashen36, jinyu6, TD10, A41772BBt, chuangmian58, mian9001, yanmin38, EZ10, xinhuimian230, xinluzhong73, TD11, xinluzhong52, TD12, zhongmiansuo16, jimian126, qianhai6, xinluzhong59, shidaK9, tada1611, TD13, mianmian3, TD14, kangchongmian5, zhongmiansuo44, xin6015, shengmian2, jimian2016, TD15, ji169, xinluzhong88, zheda304, hemian20, su1056I-1, tada1619, TD16, zhongmiansuo21371, limian12, nan6, huimin605, ruifeng2, xinzhi5, zhongmiansuo40
highly drought-sensitive	0.10~0.19	23NJH02, xiluzao12, TD17, youzhi8, huamian532, xiluzao13, jimian262, tada2, wanmian37, jiumianK58, qianhaiC, zhongmiansuo49, zhongmiansuo36, E21525-1, xiluzao65, yinmian2, TD18, emian39, Tazan-1, hemian18, changde184, TD19, xinhai23, TD20, jinken1565, jiangnanlu1, tianza26, caike586, tianyu1904, jiyou851, shandong105, xinluzao21, TD21, TD22, xinyu7, jinghuamian174, jiumianK1829, X19075, E21525-2, ZS061, su702, R8166, Ari971, ning523, jifeng197, xingjinghua206, M-8124-1159, TD23, Belshinuo, xinzamian1, lu22, xinluzao66, TD24, zhongmiansuo30, ao7, huayu708, TD25, TD26, TD27, chuanD45, taiyuan0237, TD28, huamian1543, su8908, caimianzong3, TD29, jinggang249, chuanjian3, xingjinghua231, zhongmiansuo19, haoda2861, jinfenghe8, TD30, HF52UP, lumian29, xinluzhong14, xinluzhong38, taiyuan02-41, TD31, xibeiquyu3, chuangmian509, xinluzhong56, xinluzao22, TD32, 23NJH03, damianling69, 433Bt, xinhuimian233, liaomian19, qianjinmian, xinluzhong77, TD33, huimin602, caimianlv4, xinluzhong22, xinluzhong55, jixin6, lu27, zaozhi2A, xinluzao20, jinke255, xinluzhong67, lu21, junmian1, shihezi874, chuangmian548, TD34, xinluzhong26, qianhaiA, xinluzao31, zhaofeng1, jimian16, TD35, xinluzao4

Note: D ≥ 0.40 indicates highly drought-resistant, 0.30 ≤ D < 0.40 indicates moderately drought-resistant, 0.20 ≤ D < 0.30 indicates drought-sensitive, and D < 0.20 indicates highly drought-sensitive.

**Table 6 plants-14-01770-t006:** Variance analysis of SODs under drought stress.

Source of Variation	Square	Degree of Freedom	Mean Square	F Value	*p* Value
variety type	20.91	1	20.91	53.68	0.0001
times	34.08	4	8.52	21.88	0.0001
variety type × time	59.06	4	14.77	37.91	0.0001
ERR	54.53	140	0.39		
total variation	168.57	149			

**Table 7 plants-14-01770-t007:** Variance analysis of Pro under drought stress.

Source of Variation	Square	Degree of Freedom	Mean Square	F Value	*p* Value
variety type	7118.75	1	7118.75	53.94	0.0001
times	18,524.12	4	4631.03	35.09	0.0001
variety type × time	13,659.98	4	3414.99	25.87	0.0001
ERR	18,477.93	140	131.99		
total variation	57,780.79	149			

**Table 8 plants-14-01770-t008:** Variance analysis of MDA under drought stress.

Source of Variation	Square	Degree of Freedom	Mean Square	F Value	*p* Value
variety type	1.54	1	1.54	1.56	0.2144
times	59.77	4	14.94	15.05	0.0001
variety type × time	209.23	4	52.31	52.69	0.0001
ERR	138.97	140	0.99		
total variation	409.51	149			

## Data Availability

Data are contained within the article.

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
