# Peer review of "Comprehensive Evaluation of 202 Cotton Varieties (Lines) and Their Physiological Drought Resistance Response During Seedling Stage"

_plants, 2025, doi:10.3390/plants14121770_

Round 1

Reviewer 1 Report

Comments and Suggestions for Authors

Baha’s study evaluated 202 cotton varieties’ drought resistance using several physiological traits and provided useful data. The manuscript was logically organized and wrote. It could be published after making the following revisions:

  1. The propose is that the discussion section should systematically elucidate the merits of the entropy weighting method in cotton drought resistance evaluation, particularly through comparative analysis with conventional weighting techniques such as Analytic Hierarchy Process (AHP). Comparative analysis with analogous studies in discussion section is needed.
  2. These analytical results demonstrate that plant height exhibited the highest contribution weight in both principal component analysis (PCA) and partial correlation analysis regarding drought resistance evaluation. It is recommended to provide a detailed discussion regarding the physiological basis of this phenomenon, particularly through the plant physiological mechanism.
  3. While root depth and biomass constitute critical drought-adaptive traits in conventional agronomic, their low weighting coefficients observed in this study necessitate a systematic exploration of potential drivers.
  4. The PCA result graph can be presented as a scatter plot for intuitive comparison. The ranking results of drought resistance coefficients can be attached with a cluster analysis chart to reveal the hierarchical relationship of drought resistance among varieties.
  5. It is noted that important research progress in the evaluation of crop stress resistance in recent years (such as literature after 2023) has not been cited. It is suggested that the references be updated.

Reviewer 2 Report

Comments and Suggestions for Authors

This manuscript provides a valuable and well-executed investigation into drought resistance across 202 cotton varieties at the seedling stage, using a combination of morphological and physiological indicators. The application of the entropy weight method, principal component analysis, and correlation techniques strengthens the comprehensive evaluation. The identification of elite drought-resistant lines offers significant potential for breeding and resource development. The manuscript is generally well-organized and presents important findings. However, a few revisions would enhance its clarity, structure, and overall scientific impact. Below are detailed and section-specific suggestions for improvement.

Revision Suggestions

  1. Section 1 Introduction:
  • In Lines 51-61, the explanation of the entropy weight method can be improved for clarity. For example, split the sentence starting from “The entropy method is an objective weighting method...” into two parts to avoid a run-on structure.
  • At Line 85, clearly state the objective of the study as a separate sentence: “The main objective of this study was to comprehensively evaluate drought resistance in cotton seedlings using morphological traits and physiological indices under drought stress.”
  1. Section 2.1.1 Statistical Analysis of Dc:
  • In Line 103, correct “which is 109.64%,This indicates” to: “which is 109.64%. This indicates...”
  1. Section 2.1.2 ANOVA and Correlation:
  • Line 122-123 discusses correlation results but is long and difficult to follow. Break this into 2-3 shorter sentences and specify key relationships more clearly.
  • In Figure 1, make sure all trait abbreviations are defined clearly in the caption (PH, CND, NL, etc.), even though they are previously introduced in Table 1.
  1. Section 2.1.3 PCA:
  • At Line 139, change “with a contribution rate of 38.77%, The load values” to: “with a contribution rate of 38.77%. The load values...”
  • In Table 3, consider bold the highest loadings for each principal component to visually assist readers in identifying the key traits contributing to each PC.
  1. Section 2.2 Analysis of Physiological Indicators:
  • In Line 190, clarify “with the highest 3.51 U/mg on 19d” to: “with the highest SOD activity of 3.51 U/mg recorded on day 19.”
  • In Lines 225-229, revise for clarity: the phrase “showed a trend of first increasing and then decreasing...” can be restated as: “MDA levels initially increased and then declined in drought-resistant materials, while continuing to rise in sensitive materials.”
  1. Section 3 Discussion:
  • Lines 259-263 repeat earlier results. Consider summarizing briefly and linking more directly to implications for breeding and physiology.
  • In Line 289, replace “maximum value was at 19d” with: “The highest Pro content was observed at day 19.”
  1. Section 4.3 Comprehensive Evaluation of Drought Resistance:
  • The equations from Lines 335-357 are not formatted properly for readability. Either place them in numbered display format or move to supplementary material and refer to them in-line.
  • Ensure all symbols are defined (e.g., Dc, Xij, Wj, dj, etc.), especially for international readers unfamiliar with entropy weighting calculations.
  1. Throughout the Manuscript:
  • Review text for punctuation issues, particularly comma splices and missing periods (e.g., Lines 103, 135, 190).
  • Ensure consistency in terminology. For example, “cotyledon node diameter” is sometimes referred to as “diameter of cotyledon node”, select one phrasing throughout the manuscript.
  • Consider inserting a short Conclusion section after the Discussion to summarize the key outcomes and implications for future research or breeding programs.
Comments on the Quality of English Language

The manuscript is generally understandable, and the scientific content is conveyed clearly. However, there are several instances of awkward sentence structure, inconsistent phrasing, and minor grammatical errors throughout the text. These issues are especially noticeable in the Introduction, Results, and Discussion sections. For improved clarity and fluency, a thorough language revision by a native English speaker or a professional editing service is recommended. This will ensure that the presentation matches the high quality of the scientific work.
